# Efficacy of Oil and Photosensitizer against *Frankliniella occidentalis* in Greenhouse Sweet Pepper

**DOI:** 10.3390/antibiotics12030495

**Published:** 2023-03-02

**Authors:** Zelda Pieterse, Rosemarie Buitenhuis, Jun Liu, Michael Fefer, Inna Teshler

**Affiliations:** 1University of Guelph, 4890 Victoria Avenue North, Vineland Station, ON L2R 2E0, Canada; 2Vineland Research and Innovation Centre, 4890 Victoria Avenue North, Vineland Station, ON L2R 2E0, Canada; 3Suncor AgroScience, 2489 North Sheridan Way, Mississauga, ON L5K 1A8, Canada

**Keywords:** western flower thrips, *Frankliniella occidentalis*, photodynamic inactivation, biocontrol, chlorophyllin, reactive oxygen species, integrated pest management, mineral oil

## Abstract

Many common insect pests have developed resistance against the pesticides currently available, to the point where pest and disease management has become extremely difficult and expensive, increasing pressure on agriculture and food production. There is an urgent need to explore and utilize alternatives. Due to their unique mode of action, photosensitizers may be able to control insect pests effectively, especially in combination with oil-based products, without the risk of resistance build-up. In this study, the efficacy of a mineral oil-based horticultural spray oil, PureSpray™ Green (PSG), and a sodium magnesium chlorophyllin photosensitizer formulation, SUN-D-06 PS, were evaluated and compared to a registered cyantraniliprole insecticide (as positive control) and a negative control against western flower thrips (WFT), *Frankliniella occidentalis*. In detached leaf ingestion assays, PSG at high concentration was more effective than low concentration, causing >70% WFT mortality, whilst SUN-D-06 PS + PSG caused higher mortality than cyantraniliprole after five days of feeding. The same combination was as effective as cyantraniliprole in the contact assay. In greenhouse pepper, the photosensitizer decreased the WFT more than mineral oil applied alone, whilst a combination treatment of SUN-D-06 PS + PSG was most effective, decreasing the WFT population to fewer than four WFT per plant. SUN-D-06 PS + PSG shows promise as a sustainable, economical way of controlling WFT, with the potential to be incorporated into existing integrated pest (and disease) management (IPM) programs with ease.

## 1. Introduction

Western Flower Thrips (WFT), *Frankliniella occidentalis* (Pergande) (Thysanoptera: Thripidae), are one of the most invasive, destructive insect pests of horticultural crops worldwide, having already spread to at least 57 countries and continuing as the result of international trade [1,2]. WFT are a polyphagous pest with both soil-dwelling developmental stages (late second larval instars, pre-pupae, and pupae) and foliar-feeding stages (adult, first and second larval instars) that cause direct damage to plants by feeding on leaf, flower and fruit surfaces, and indirect damage by vectoring viruses [3,4]. More than 80% of plant diseases are transmitted by insect vectors [5], with WFT being the primary vector for Impatiens Necrotic Spot Virus as well as Tomato Spotted Wilt Virus and having the ability to spread Orthotospoviruses, Ilarviruses, Alphacarmoviruses and Machlomoviruses [2,6]. Park et al. [7] reported that the economically tolerable ratio for damaged pepper fruits is less than 8%, while in many high-value ornamentals there is zero tolerance for any damage; as a result, the threshold for WFT is near zero [4,8].

WFT are notoriously difficult to control due to their rapid population increase, cryptic behaviour, high level of vagility and polyphagous nature [2,3,9]. Additionally, WFT have a propensity for developing insecticide resistance, primarily associated with four biological parameters: (a) short generation time, (b) high female reproductive capacity (fecundity), (c) a haplo-diploid breeding system and (d) their diverse detoxification enzymes, including cytochrome P-450 monooxygenases, esterases, and glutathione S-transferases [5,10]. The Arthropod Pesticide Resistance Database (www.pesticideresistance.org) lists at least 175 documented cases of insecticide resistance in WFT to at least 30 active ingredients. This includes at least ten chemical classes with varying modes of action, including traditionally used broad spectrum insecticides, such as pyrethroids, neonicotinoids, organophosphates and carbamates, as well as more recently developed ‘reduced risk’ chemistries such as spinosad [2,4,11]. The use of insecticides may also have unwanted side-effects, such as the mortality of natural enemies (predators) and secondary pest outbreaks, and ultimately provided WFT with a competitive advantage over other species [2].

Due to these factors and supported by the activities of researchers and extension specialists, a complete paradigm shift has occurred within the last two decades in Canada, with biological control (the use of natural enemies, including parasitoids and predators) now at the core of thrips management, especially in greenhouses [4,12]. It is usually necessary to use several different biocontrol agents and strategies to control WFT as part of an IPM program, as a single biocontrol agent does not provide adequate control, especially when pest pressure is high [4,8]. Despite the success of biological controls of thrips, intervention with pesticides is still needed occasionally, for example to control a secondary pest, to reduce pest populations to levels where they can be controlled biologically, to treat pest hotspots or as a clean-up spray. It is in these circumstances that the need to explore alternative products that can be used without negative effects on biocontrol agents as part of a comprehensive integrated pest (and disease) management (IPM) program becomes critical. This study investigates two types of reduced risk products, mineral oils and photosensitizers, that both have the potential to fill this gap, either alone or in combination.

Horticultural (mineral) oils are compatible with modern, sustainable agriculture practices, including IPM, and pose several advantages over conventional pesticides such as very low mammalian toxicity, low residual activity and significantly decreased disruption to biological control programs compared to broad-spectrum insecticides [10,13]. Although petroleum-based mineral oils have been used for insect pest control for over a century, they have only gained popularity for regular use as fewer options have become available in recent years, with continuing pest resistance to conventional products [10,14].

Studies on the efficacy and chemistry of petroleum-based oils over the last sixty years led to the identification of the main factors related to their insecticidal activity as well as their potential phytotoxicity [14]. Although popular opinion concurs that insect suffocation by spiracle blockage is usually the mode of action, oils show high affinity to the insect body surface and penetrate the insect cuticle, dissolve internal lipids and penetrate internal cell structures [15,16]. Oils also cause the inhibition of water excretion, cell dehydration, DNA condensation and further physiological stress, which leads to insect death either through the disruption of internal organs or due to prolonged immobilization [17]. No resistance of any insect pest to oil has been reported, possibly due to their effects on multiple targets as opposed to single sites. Oil is lethal to insect pests due to the sum of its effects, whether it be breakdown of cuticle waxes, cuticle softening, epidermal teratogenicity, tracheal blockage, receptor coating, deterrence and neurotoxicity [13].

Due to their unique mode of action, photosensitizers could also play an important role in IPM as reduced risk pesticides. Recently, Photodynamic Inactivation (PDI) has emerged as a powerful tool to control both Gram-negative and Gram-positive bacteria, as well as fungi [18,19]. PDI is characterized by the use of a photosensitizer (PS), that when activated by a light source in the presence of molecular oxygen produces reactive oxygen species (ROS) and singlet oxygen. These reactions target multiple sites, causing extensive damage leading to rapid cell death and subsequent insect death (Figure 1). To date, no known resistance to PDI has been recorded [19]. It is highly unlikely that any organism could develop resistance against photosensitizers since their mode of action targets multiple sites at the same time and they become exhausted after light-activation.

Although PDI is believed to have been accidentally discovered in 1900 when Oscar Raab incubated microorganisms with certain dyes in both light and dark conditions, it has been more extensively studied for use in human medicine (such as in cancer treatment, for example) than in any other field to date [18,20]. The ideal structure of a PS used in horticulture differs considerably from its counterpart used in human medicine. Whilst repeated medical photodynamic therapy may make patients sensitive to light, photosensitizers as applied in horticulture are safe to use, not cytotoxic, nor genotoxic or mutagenic to human and animal life [19,20]. Indeed, the sodium magnesium chlorophyllin (Chl, E140) photosensitizer formulation evaluated in this study (Figure 2) is derived from natural molecules, and has been widely approved as food additive E140, “natural green”. Referred to as ‘pigments of life’, Xanthenes and porphyrins (including chlorophyllins, such as Chl, E140) appear to work best in biological applications due to their high photo-insecticidal activity, water-soluble nature and quick photodegradation, preventing a buildup of phytotoxicity [21].

Both mineral oil and photosensitizers can be used alone, or can be combined. If combined, the concentration of mineral oil may potentially be reduced in order to mitigate barriers to their use, such as phytotoxicity. The objectives of this study were to evaluate the efficacy of a registered horticultural petroleum-based mineral oil, PureSpray™ Green (PSG), and a newly developed chlorophyllin-based photosensitizer formulation, SUN-D-06 PS, alone or in combination, against WFT in detached leaf assays under different light conditions and in greenhouse trials. To our knowledge, this is the first study reporting on the use of photosensitizer against WFT.

## 2. Results

### 2.1. Detached Leaf Assays

#### 2.1.1. Ingestion Assay

In the ingestion assay, WFT did not come into direct contact with treatments and were only exposed by ingesting treated plant material. There was a low natural mortality (less than 15%) of WFT feeding on cabbage leaf discs sprayed with water (negative control) throughout the trial, with a slight but significant increase from day 2 to day 5 (Figure 3). In the oil treatments, the high concentration (PSG 1%) caused a higher mortality of WFT than the low concentration (PSG 0.25%). Adding the low concentration of oil to the PS treatment increased WFT mortality compared to the PS or the oil at 0.25% alone (F_(5,574)_ = 380.07, *p* < 0.0001).

The different lighting conditions, minimal vs. LED light, did not affect WFT mortality in the reverse osmosis (RO) water, mineral oil (PSG) 1% and the photosensitizer SUN-D-06 PS treatments. There was a significant interaction between treatment and light for WFT mortality observed with PSG 0.25%, cyantraniliprole and SUN-D-06 PS 0.22% + PSG 0.25% (F_(5,574)_ = 31.15, *p* < 0.0001).

There was a significant interaction between treatment and time, with WFT mortality observed for PSG 0.25%, PSG 1%, SUN-D-06 PS 0.22% and SUN-D-06 PS 0.22% + PSG 0.25% under minimal light, while the same was true for SUN-D-06 PS 0.22% and SUN-D-06 PS 0.22% + PSG 0.25% under LED light conditions (F_(6,568)_ = 9.18, *p* < 0.0001).

Overall, the best performing treatment compared to the positive control (cyantraniliprole) was the treatment combining the photosensitizer with oil at a low concentration, SUN-D-06 PS 0.22% + PSG 0.25%, under LED light conditions. This treatment caused WFT mortality similar to the positive control after two days of ingesting treated plant material, whilst causing higher mortality than cyantraniliprole after five days of feeding.

#### 2.1.2. Contact Assay

In the contact assay, WFT came into direct contact with treatments and subsequently fed on treated plant material. There was low natural mortality (less than 15%) of WFT throughout the trial (Figure 4). There was no significant interaction between treatment and light for any of the treatments evaluated (F_(5,207)_ = 165.87, *p* = 0.2377). There was no significant interaction between treatment and time for any of the treatments (F_(5,207)_ = 124.80, *p* = 0.4858). Therefore, the full effect of the photosensitizer can already be observed two days after treatment. The higher concentration of mineral oil, PSG 1%, caused significantly higher WFT mortality compared to the low concentration PSG 0.25% (F_(5,167)_ = 183.36, *p* < 0.0001). The combination of the photosensitizer with a low concentration of oil, SUN-D-06 PS 0.22% + PSG 0.25%, was as effective at causing WFT mortality as the high concentration of registered oil product, PSG 1%, and the positive control, cyantraniliprole. When used in combination with oil at a low concentration the photosensitizer was as effective as the positive control in both minimal and LED light conditions (Figure 4).

### 2.2. Greenhouse Assay

No damage due to phytotoxicity of any of the treatments was found in any of the plants throughout the trial.

All treatments, the mineral oil (PSG 0.25%), the photosensitizer, SUN-D-06 PS 0.22%, and the combination of SUN-D-06 PS 0.22% + PSG 0.25%, significantly decreased the WFT populations in bell pepper grown in insect cages under commercial greenhouse conditions, compared to the negative RO water control (F_(3,359)_ = 162.274, *p* < 0.0001). In the negative control treatment, the WFT population increased from ~5 WFT to >50 WFT per plant in a three-week period (Figure 5A).

The decrease in the WFT population observed on pepper plants was similar for PSG 0.25% and PSG 1%, indicating that PSG applied at a lower than recommended label rate concentration controlled WFT under the conditions of this trial (Figure 5A). Mineral oil decreased both larval and adult stages of the WFT population (Figure 5B,C).

The photosensitizer formulation, SUN-D-06 PS 0.22%, decreased the WFT population significantly more than mineral oil applied alone. After the pepper plants were infested with ~5 adult WFT per plant, SUN-D-06 PS effectively controlled the population with fewer than 8 WFT (of any stage) per plant observed at any time during the trial (Figure 5A). The photosensitizer, SUN-D-06 PS 0.22%, was effective in controlling both larval and adult stages of the WFT population (Figure 5 B,C).

Adding mineral oil at a low concentration to the photosensitizer treatment did not have a significant effect in further decreasing the larval population of WFT (Figure 5B). In contrast, a significant interaction was observed with the aforementioned combination, SUN-D-06 PS 0.22% + PSG 0.25% and the adult stage of WFT (Figure 5C). Initially, there was no significant difference in the number of adult WFT observed on plants treated with SUN-D-06 PS 0.22% compared to SUN-D-06 PS 0.22% + PSG 0.25% during the first two weeks of the trial. However, by week 3 a larger decrease in adult WFT treated with SUN-D-06 PS 0.22% + PSG 0.25% was observed, indicating that the addition of a low concentration of oil has a significant, albeit delayed effect. Looking at data combining both larval and adult stages of WFT (Figure 5A), a significant interaction is again observed at week 3, with SUN-D-06 PS 0.22% + PSG 0.25% causing a larger decrease in total WFT population compared to SUN-D-06 PS 0.22% applied alone. Overall, SUN-D-06 PS 0.22% + PSG 0.25% most effectively controlled the WFT population, with fewer than four WFT per plant being observed on pepper plants treated with this photosensitizer–organic oil combination after three weeks.

## 3. Discussion

There is an urgent, critical need for sustainable, environmentally conscious products to control insect pests and diseases worldwide. Insects such as WFT, a pest and vector notorious for developing resistance against insecticides, have become impossible to control without the use of comprehensive integrated pest and disease management (IPM) strategies. Novel products and strategies are needed to control WFT that are compatible with current biological control-based IPM programs.

Given the long history of using mineral oils for managing insect pests, as well as the efficacy with which spray oils cause rapid death in smaller insects [13,17], we hypothesized that PureSpray™ Green oil (PSG) would be effective in causing the mortality of WFT when sprayed. Indeed, PSG was effective in controlling WFT in the ingestion, contact and greenhouse assays in this study. More recently, Durr et al. [10] observed that essential oils of peppermint, fir, arborvitae and thyme were effective in reducing a greenhouse WFT population relative to a water control. We hypothesize that mineral oil causes a variety of effects, including “arrested activity” in WFT with a repellent effect that discourages egg disposition and feeding, consequently making it effective even at low concentration, as found in this study. Interestingly, we observed an increased efficacy of the photosensitizer, SUN-D-06 PS, when used in combination with a low concentration of PSG. One possible explanation is that, given their lipophilic nature, oil accumulates in cell membranes and thus affects their structural and functional properties. This action may further limit thrips’ ability to rid itself of both the PSG and SUN-D-06 PS, increasing the thrips’ vulnerability to the effect of the photosensitizer in the current study. A similar finding was recorded by Najar-Rodríguez et al. [17] in their study on the efficacy of oil on the cotton aphid, *Aphis gossypii* Glover. They alluded to the possibility that, in addition to causing anoxia, oil penetrates the insect cuticle and the internal cell structure. Buteler and Stadler [13] confirmed the different modes of action of oil, including the inhibition of excretion.

Throughout our trials, mineral oil was as effective as cyantraniliprole in causing the mortality of WFT. Cyantraniliprole is an anthranilic diamide acting on the ryanodine receptors in insect muscle cells. Cyantraniliprole has proven to be toxic to larvae and adult WFT [22,23]. In the ingestion assay in this study, there was a significant interaction between cyantraniliprole and the light condition; we hypothesize that the muscular contractions and paralysis associated with cyantraniliprole, combined with the intense light, discouraged the feeding of WFT, whereas in the contact assay there was no difference due to the full effect of cyantraniliprole occurring at contact. Cyantraniliprole was originally popular due to its ability to provide cross-spectrum control of chewing and sucking pests whilst being less toxic (classified as class 2, slightly harmful, according to the definitions of toxicity given by the International Organization for Biological Control (IOBC)) to biocontrol agents, such as *Orius insidiosus* (Say) (Hemiptera: Anthocoridae) [22,23]. Cyantraniliprole is a registered pesticide in Canada and the US, with no WFT resistance reported to date, although Wang et al. [24] have reported WFT resistance to cyantraniliprole in China. WFT use detoxification enzymes including cytochrome P-450 monooxygenases, and P450s have been confirmed as a key factor in the initial development of cyantraniliprole resistance by *Aphis gossypii* Glover and field-evolved resistance in *Bemisia tabaci* (Gennadius) (Hemiptera: Aleyrodidae) [10,25,26]. It is therefore possible that thrips’ resistance to cyantraniliprole will continue to increase and spread geographically, intensifying the need for alternatives.

The photosensitizer, SUN-D-06 PS, was compared to cyantraniliprole as a “commercially accepted control” for thrips management, as required for registration in Canada. While SUN-D-06 PS caused WFT mortality > 50%, effectively suppressing the pest, the combination treatment of SUN-D06 PS + PSG was as effective as cyantraniliprole in causing WFT mortality > 75% in the contact assay. Even when WFT did not come into direct contact with the treatments, as in the ingestion assay, SUN-D-PS with PSG still caused WFT mortality > 70% and was more effective than cyantraniliprole under LED light conditions (day 5 observations).

We observed no difference in WFT mortality in both the contact and the ingestion assays for SUN-D-06 in minimal light vs. LED light. Only in the ingestion assay was there a significant increase in mortality in LED light conditions compared to minimal light with the combination treatment of SUN-D-PS + PSG. These results were unexpected, as Glueck et al. [18] found that sodium magnesium chlorophyllin (Chl, E140) did not show any dark toxicity. In their studies on bacteria, light was required to activate the photosensitizer and kill *Rhodococcus fascians*, *Xanthomonas axonopodis* and *Erwinia amylovora*. Additionally, no dark toxicity was observed for Chl, at concentrations required to kill insects, with light a critical requirement to activate the photosensitizer against the camel tick *Hyalomma dromedarii* (Koch) [27]. It should be noted that we did not include a completely dark control because the WFT did not survive or feed well without any light. Preliminary trials for this study showed that the photosynthetically active ingredient in SUN-D-06 caused significantly higher mortality of WFT in LED light conditions compared to minimal light conditions. It is possible that the formulated product is more reactive to light under similar conditions, or that the light conditions in the growth chamber may have inadvertently changed.

The results of the detached leaf assays were validated in the greenhouse trial. Here, the difference between SUN-D-06 PS and SUN-D-06 PS + PSG was not as pronounced as in the detached leaf assays. Still, SUN-D-PS decreased the WFT larval population by 85% and the WFT adult population by 76% compared to the water control after only one spray. This excellent control was maintained throughout the three-week observation period, with an 88% decrease in WFT population compared to the control at the end of the trial, whilst SUN-D-06 PS + PSG controlled the WFT population the most effectively, with a 93% decrease in WFT population. In other greenhouse studies, pyridalyl, chlorfenapyr, spinosad and spinetoram provided 80, 90, 94 and 96% control, respectively [28,29], whilst specifically in pepper, Srivastava et al. [30] found spinetoram controlled adult and larval WFT thrips between 32 and 83% and 93% six days after application. In comparison, cyantraniliprole was observed to reduce the WFT population in greenhouse pepper by only 60% compared to a water control, and was considered as moderately effective compared to spinetoram [30]. These comparisons highlight the efficacy of the photosensitizer, SUN-D-06 PS. The economic threshold of WFT in greenhouse grown peppers was found to be 0.7 to 2.1 WFT per flower. With each plant having between 5 and 10 flowers, a threshold range of 3.5 to 21 WFT per plant is acceptable [7]. In the greenhouse assay in this study, there were only ~4.6 WFT observed per pepper plant after a single spray with SUN-D-06 PS, whilst SUN-D-06 PS both as a stand-alone product and in combination with mineral oil maintained WFT control throughout our trial with WFT numbers continuously below seven per plant.

Photosensitizers have been observed as effective against bacteria [18], fungi [19] and insects such as mosquitoes, ticks and *Thrips tabaci* Lindeman [31,32]. Glueck et al. [18] reported that the sodium salt of chlorophyllin (Chl, E140), the active ingredient present in SUN-D-06 PS in this study, is effective in controlling the Gram-positive bacterial plant pathogen *Rhodocccus fascians* and Gram-negative *Xanthomonas axonopodis* and *Erwinia amylovora*. PDI using Chl was also effective in photo-killing *E. amylovora* resistant to the antibiotic streptomycin [33]. Hamminger et al. [19] observed that the Chl photosensitizer, when activated by LED light, effectively inhibited the mycelial growth of *Alternaria solani* and *Botrytis cinerea*. Additionally, Chl successfully photokilled and eradicated both fungi without causing any phytotoxic symptoms in *Fragaria vesca*. This means that SUN-D-06 PS potentially has efficacy to control multiple pests and pathogens, including those that display resistance to other treatments, in greenhouse crops.

To ensure a good fit with IPM programs using beneficial insects and mites, it would be required that photosensitizers are effective against insect pests but not against biological control agents and other beneficial insects. Some studies indicate that photosensitizers have been successfully employed against larger insects. For example, photodynamic processes using rose-bengal photosensitizers have been recorded to be 100 times more effective than the commercially available chlorpyrifos insecticide and effective against pesticide-resistant mosquitoes [32,34]. However, SUN-D-06 PS as a chlorin-based photosensitizer should not pose any threat to pollinators. Nassar and El-Tayeb [35] observed that the maximum mortality rate in bees did not exceed 10%, even with an increased concentration of 7 × 10^−6^ M/L of chlorophyllin applied. Bees showed a high efficiency in expelling accumulated chlorophyllin derivates within 48 h [34].

In IPM programs for WFT management, compatibility with biocontrol agents such as the minute pirate bug *O. insidiosus* is important. *Orius* spp. are generalist predators, with both nymphs and adults feeding on WFT larvae and adults [36]. *Orius* spp. have been successfully used as biological control agents against WFT in sweet pepper for several years in different geographical locations [36,37,38]. Funderburk et al. [36] recorded a near-extinction of WFT adults and larvae once *O. insidiosus* prey ratios reached 1:40. Pieterse et al. (unpublished data) have observed that *O. insidiosus* can be used with the photosensitizer SUN-D-06 PS to control WFT. SUN-D-06 PS did not cause mortality in *O. insidiosus*, showing promise as a product to add to the IPM toolbox.

The availability, price, water-solubility and biocompatibility of PS are all critical factors that will determine its success in commercial markets. Since sodium magnesium chlorophyllin (Chl, E140) can be sourced from cyanobacteria and blue-green algae, it is readily available and economical. It is also water-soluble and compatible with oil-based products and potentially compatible with many biological control agents. Therefore, SUN-D-06 PS when applied with PSG should allow for the sustainable, economical control of WFT in greenhouses and have the potential to be incorporated into existing IPM programs with ease. Future research may include confirming the efficacy of SUN-D-06 PS against other important insect pests. Additionally, confirming the compatibility of SUN-D-06 PS with biological control agents, including predators and parasitoids currently being used, as well as beneficials, such as pollinators, is of importance.

## 4. Materials and Methods

### 4.1. Western Flower Thrips Colony

A WFT colony was established from individuals collected from roses at the Vineland Research and Innovation Centre, Vineland Station, ON, Canada. A mixed-aged colony was subsequently reared and maintained on marigold plants (*Tagetes patula* ‘Bonanza yellow’) and bean plants (*Phaseolus vulgaris* ‘California red kidney’) in thrips-proof screened dome-shaped cages (BugDorm—2120F, MegaView Science Co., Ltd., Taichung, Taiwan) inside a growth chamber (23 ± 1 °C, 60% RH, and 16:8 L:D). Individuals were collected directly from the cages using an aspirator when required.

### 4.2. Treatments

In all assays, reverse osmosis (RO) water was used as a negative control. In the detached leaf assays, cyantraniliprole (Exirel^®^, iFMC of Canada Ltd., Mississuaga, ON, Canada) applied at recommended label rate was used as a positive control. Cyantraniliprole is a group 28 insecticide, a member of the anthranilic diamide class, with cyantraniliprole 100 g/L as the active ingredient. Other treatments evaluated included PureSpray™ Green Spray Oil 13E (PSG) (Intelligro™, Suncor Energy Inc., Mississauga, ON, Canada) and the photosensitizer. PSG is an OMRI listed, organic, broad-spectrum, emulsifiable horticultural spray oil with 99% mineral oil as the agricultural active ingredient. The photosensitizer evaluated was a pre-commercial formulation called SUN-D-06 PS (Suncor AgroScience, Suncor Energy Inc., Mississauga, ON, Canada). When used at the recommended rate of 0.22% it contains 0.88 µM Magnesium Chlorophyllin as the photosensitizer agent and 6.6 µM Disodium EDTA (Ethylenediaminetetraacetic acid) as a chelating agent to enhance the penetration. Both these compounds are safe, food grade chemicals that do not cause phytotoxicity to plants. SUN-D-06 PS can be tank mixed with low-rate organic pesticides, hence a treatment combining SUN-D-06 PS with PSG was also evaluated. The photosensitizer formulation was stored at 4 °C in the dark until use, while all other products were stored according to recommended procedures on the label. Treatment solutions were freshly prepared before each trial with RO water in all cases.

### 4.3. Detached Leaf Assay

Two different detached leaf assays were conducted: ingestion and contact.

For ingestion assays, WFT were siphoned into insect assay containers after spraying was complete and plant material sufficiently dried. There was no direct contact of WFT with the treatment at the time of spraying, and any effects observed were due to feeding and ingesting treated plant material.For contact assays, WFT were siphoned into insect assay containers before the spraying of treatments commenced. WFT were in direct contact with the sprayed treatment and subsequently ingested the treated plant material.

Detached leaf assays were completed using a modified form of version 3.4. of method 19 of the Insecticide Resistance Action Committee (IRAC). Cabbage (variety: Charmant) *Brassica oleracea* var. *capitata* was grown from seed (West Coast Seeds, BC, Canada). Cabbage leaf discs were prepared by using a leaf punch (sharpened, sterilized metal tube) to create 20 mm Ø round circles from young (two- to four-week-old) cabbage leaves. Agar was prepared by mixing 2% *w*/*w* agar powder with reverse osmosis water, autoclaving for 15 min at 121 °C and cooling the agar solution to ~50 °C while constantly mixing on a magnetic stirrer hotplate. Insect assay arenas were prepared by pouring the prepared agar into the bases of small, translucent, shallow plastic deli cups (opening 60 mm diameter, bottom 40 mm diameter, height 30 mm, volume 2 fl. oz (Solo^®^ P200N, Dart Container Corporation, Mason, MI, USA) to a depth of 8–10 mm, allowing for at least 10 mm space between the top of the agar and the lid of the insect assay container. Prepared cabbage leaf discs were placed, abaxial side up, in the centre of insect assay containers on top of the agar (using forceps) just before solidifying. Insect assay containers were closed with vented, thrips proof lids (round vent holes of 20 mm Ø were made in the lids of deli cups and covered with 25 mm Ø thrips-proof mesh screen). Treatments (Table 1) were prepared 1–4 h before application in a spray hood in minimal light conditions. Treatments were contained in dark (black, non-translucent) hand-held spray bottles (travelagn, Amazon.ca) to prevent deterioration of photosensitive compounds. From each treatment, 0.2 mL was sprayed onto each leaf disc in insect assay containers using fine mist spray nozzles (with standard fan-shaped distribution pattern and average particle size less than 100 microns). The ingestion assay was completed in 5 blocks (individual trials), each including the negative control, RO water. Not all treatments were evaluated in each block due to limitations of time, space and availability of containers. In total, 80 replicates were assessed for the negative control while at least 40 replicates were assessed for all other treatments (Table 1).

For ingestion assays, leaf discs were allowed to air dry in darkness in the spray hood after spraying before introducing WFT. Ten adult WFT per assay container were siphoned onto each leaf disc, after which the vented lids were secured.

For contact assays, ten adult WFT were siphoned onto each leaf disc before spraying. For treatment application, vented lids were momentarily opened and 0.2 mL of the respective treatments (Table 2) were sprayed onto leaf discs with WFT present, in insect assay containers using the same equipment and technique as in the ingestion assay. After spraying, vented lids were secured and leaf discs were allowed to air dry in darkness, inside the spray hood. The contact assay was completed in 4 blocks (individual trials), each including the negative control, RO water. Not all treatments were evaluated in each block, due to limitations of time, space and availability of containers. In total, 56 replicates were assessed for the negative control, while at least 16 replicates were assessed for all other treatments (Table 2).

For both ingestion and contact assays, insect assay containers were placed under either high intensity light emitting diode light (LED, at an average of 500 μmol.m^−2^s^−1^ PAR (photosynthetically active radiation)) or under low intensity incandescent light (minimal light, at an average of 50 μmol.m^−2^s^−1^ PAR) in a Conviron chamber (Controlled Environments Limited, Conviron Canada, Winnepeg, MB, Canada). Containers were placed in a randomized complete block design in all cases. Environmental conditions were 25 ± 2 °C; RH = 70% ± 5% with a 12L:12D photoperiod. Mortality (number of dead WFT) was recorded on day 2 and day 5 after spraying. Insect assay containers were placed in cold (4 °C) conditions for ten minutes to minimize the movement of WFT. WFT were counted as dead when no movement was observed initially or after being prodded with a small lab paintbrush under magnification with a stereo microscope. Data were entered into the Agriculture Research Management (ARM) software program (ARM6 Revision 2022.5) (GDM Solutions, Inc., Brookings, SD, USA). Data were analyzed using ARM recommended assessment review methods. ARM recommended square-root transformation of data in all cases where data did not meet assumptions of normality or homogeneity of residuals. To confirm the results by the ARM software, data were also analyzed by repeated measures ANOVA using the Generalized linear mixed model (Proc GLIMMIX) in SAS release 9.4 (SAS Institute Inc., Cary, NC, USA). Tukey’s multiple comparison (Tukey’s HSD test, *p* < 0.05) was used to contrast the results.

### 4.4. Greenhouse Assay

Greenhouse trials were conducted from April to December 2022 in a small research greenhouse (8.5 × 7 m) at Vineland Research and Innovation Centre, Vineland Station, ON, Canada. Snackabelle Red mini peppers (*Capsicum annuum*) (certified, untreated seed) (Stokes Seeds, Thorold, ON, Canada) were seeded in Rockwool plug trays (Grodan, Milton, ON, Canada) saturated first with clean water and then with 200 ppm 17-5-17 NPK complete fertilizer solution (Master Plant-Prod Inc., Brampton, ON, Canada) adjusted to a pH of between 5.5 and 6. The surface of Rockwool was completely covered with fertilizer-presoaked vermiculite (Uline, Milton, ON, Canada). Rockwool trays were placed in a germination chamber with temperature at 25–26 °C. At the first sign of germination the temperature was lowered to 23 °C. Seedlings were moved to the greenhouse compartment around 14 days after seeding, when the first true leaves emerged. Rockwool plugs were transplanted into pre-saturated, pre-filled 4′ pots with soilless media (Agro Mix^®^ G7, Fafard, Saint-Bonaventure, QC, Canada). Pots were placed into 60 × 60 × 60 cm dome-shaped thrips-proof (mesh: 160 μm) insect rearing cages (BugDorm—2120F, MegaView Science Co., Ltd., Taichung, Taiwan) to create the experimental arena. Nine pots were placed into each cage and cages were set up in a randomized complete block design. A drip-irrigation emitter was put in each of the pots and the plants were watered with 200 ppm 17-5-17 NPK complete fertilizer solution (Master Plant-Prod Inc., Brampton, ON, Canada) daily throughout the duration of the experiment. A 14L:10D photoperiod was followed from 6:00 a.m. to 20:00 p.m., with sunlight supplemented with high pressure sodium (hps) lighting at 180 µmol, when the light intensity was below 250 µmol. Daytime/nighttime temperatures in the greenhouse were 21–23 °C and 20 °C, respectively. Seedlings were allowed to adjust to greenhouse conditions for about 48 h, before being exposed to thrips. During this time, seedlings were dusted with pollen, Nutrimite Predatory mite food (Plant Products, Leamington, ON, Canada) using a make-up brush. Two days after transplanting, cages were infested by siphoning 50 adult, female WFT into each cage (~5.5 WFT per plant). Thrips were allowed to establish and build a population for seven days. 

Nine days after transplant (DAT) (or seven days/one week after infestation) plants inside insect cages were sprayed with their respective treatments (Table 3), using the same fine mist nozzle dark spray bottles that were used in the detached leaf trial. Plants were sprayed mid–late afternoon to the point of run-off, ensuring that both abaxial and adaxial sides of the leaves were completely covered. On the morning after spraying, three seedlings were collected from each cage and single seedlings placed into marked Ziploc bags. Ziploc bags with seedlings were placed in the fridge to slow the movement of WFT. The six plants remaining per cage were dusted with pollen (as previously described). Ziploc bags with seedlings were kept in the fridge for 1–5 days. During this time, bags were removed for counting of WFT. Each seedling was removed from its Ziploc bag and WFT adults and larvae per plant were counted directly from the leaves and other plant parts of each pepper seedling using 4× magnification goggles. WFT that remained inside the Ziploc bag at the time of removing the seedling from the bag were counted and added to the total. Seven days after the first spray (two weeks after infestation, 16 DAT) the second spray was completed as described before. The morning after the second spray three seedlings per cage were removed and the number of WFT determined as described before. Seven days after the second spray (three weeks after infestation, 23 DAT) the third (and final) spray was completed. The morning after the second spray three seedlings per cage were removed and the number of WFT determined as described before. 

For each trial there were 5 replicates per treatment, for a total of 30 cages/270 seedlings per trial. Not all treatments were evaluated in each block due to limitations of time and availability of space. In total, 90 replicates were assessed for the negative control while at least 30 replicates were assessed for all other treatments (Table 3). Data were entered into the Agriculture Research Management (ARM) software program (ARM6 Revision 2022.5) (GDM Solutions, Inc., Brookings, SD, USA). Data were analyzed using ARM recommended assessment review methods. ARM recommended square-root transformation of data in all cases where data did not meet assumptions of normality or homogeneity of residuals. To confirm the results by the ARM software, data were then analyzed by repeated measures ANOVA using the Generalized linear mixed model (Proc GLIM-MIX) in SAS release 9.4 (SAS Institute Inc., Cary, NC, USA). Tukey’s multiple comparison (Tukey’s HSD test, *p* < 0.05) was used to contrast the results.

## 5. Conclusions

With global challenges such as climate change and food security becoming ever more critical, there has been a much-needed shift towards environmentally responsible, sustainable agriculture and food production practices in recent years. Many common insect pests have developed resistance against pesticides, with *Frankliniella occidentalis* being one of the most widespread and damaging to horticulture. Alternatives such as photosensitizers may be able to control insect pests effectively, especially in combination with oil-based products, without the risk of resistance developing. This study demonstrated that, indeed, the photosensitizer SUN-D-06 PS decreased the WFT population to below threshold in this study on greenhouse pepper.

## Figures and Tables

**Figure 1 antibiotics-12-00495-f001:**
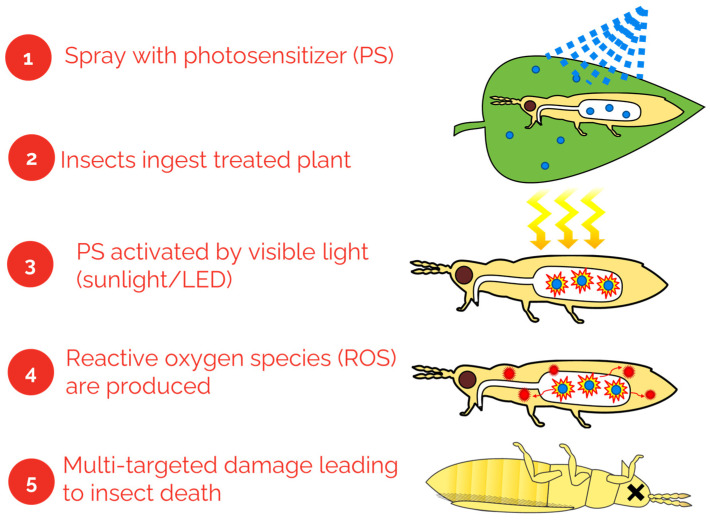
Diagrammatic illustration of mode of action of sodium magnesium chlorophyllin (Chl, E140) photosensitizer, SUN-D-06 PS, on western flower thrips (WFT), *Frankliniella occidentalis*. Graphic design and copyright: Ashley Summerfield.

**Figure 2 antibiotics-12-00495-f002:**
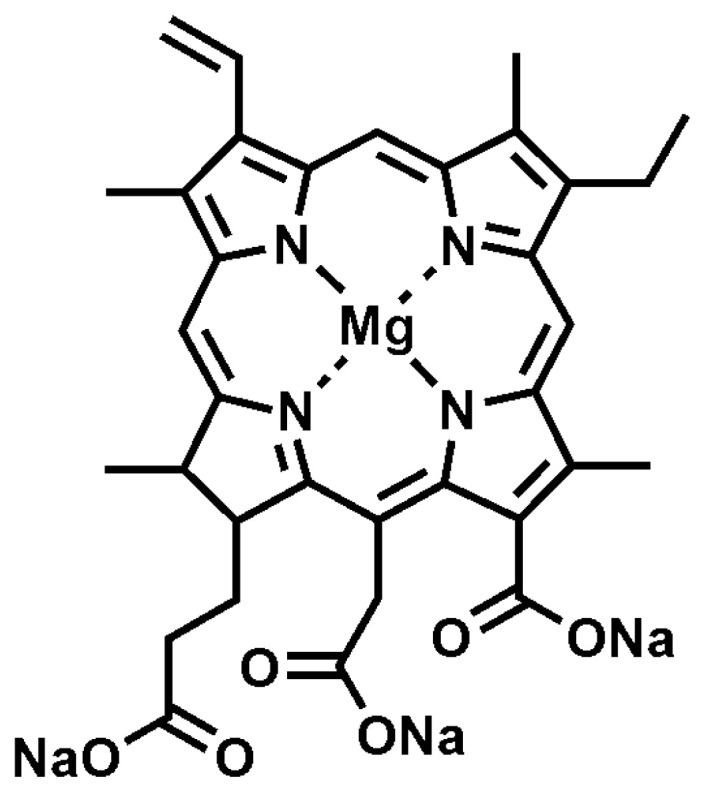
Molecular structure of Sodium Magnesium Chlorophyllin in photosensitizer formulation, SUN-D-06 PS.

**Figure 3 antibiotics-12-00495-f003:**
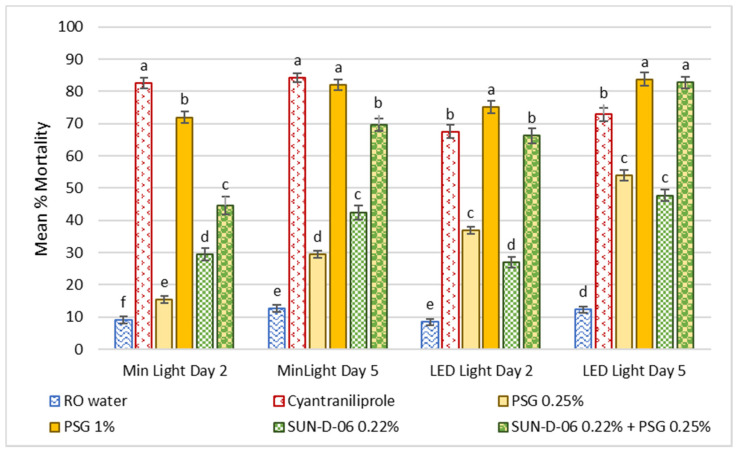
Mean percentage mortality of western flower (WFT) after feeding on treated cabbage leaf discs for two and five days, respectively, in the ingestion assay. Error bars denote standard error, and bars with the same letters indicate treatments that were not significantly different (*p* > 0.05).

**Figure 4 antibiotics-12-00495-f004:**
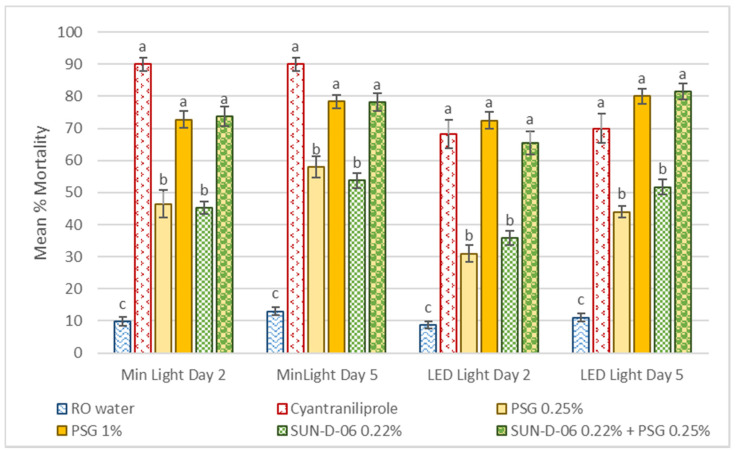
Mean percentage mortality of western flower (WFT) after feeding on treated cabbage leaf discs for two and five days, respectively, in the contact assay. Error bars denote standard error, and bars with the same letters indicate treatments that were not significantly different (*p* > 0.05).

**Figure 5 antibiotics-12-00495-f005:**
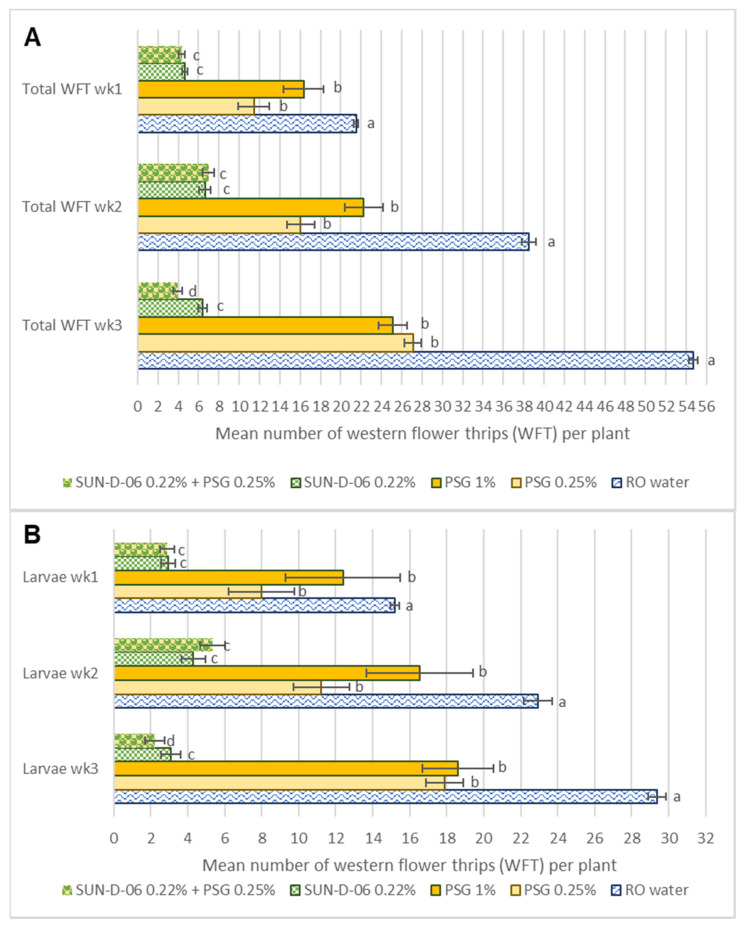
Mean number of western flower thrips (WFT) observed in weeks 1, 2 and 3 on pepper plants (seven days after infestation, week 1, 14 days after infestation, week 2 and 21 days after infestation, week 3). (**A**)—Total WFT (all stages combined); (**B**)—WFT larvae; (**C**)—WFT adults. Error bars denote standard error, while bars with the same letters indicate treatments that do not significantly differ (*p* > 0.05).

**Table 1 antibiotics-12-00495-t001:** Treatments evaluated for efficacy against western flower thrips (WFT) in ingestion assays conducted during 2021–2022. ‘x’ indicates the specific treatment was evaluated in the assay conducted during the month/year the column represents. ‘n’ indicates the total number of replicates for each treatment.

Treatments	Aug 2021	Sep 2021	May 2022	Jun 2022	Jul 2022	n = __
RO water	x	x	x	x	x	80
Cyantraniliprole	x	x	x			40
Mineral oil (PSG) 0.25%		x	x	x	x	70
Mineral oil (PSG) 1%	x		x	x	x	60
SUN-D-06 PS 0.22%			x	x	x	50
SUN-D-06 PS 0.22% + PSG 0.25%			x	x	x	50

**Table 2 antibiotics-12-00495-t002:** Treatments evaluated for efficacy WFT in contact assays conducted during 2021–2022. ‘x’ indicates the specific treatment was evaluated in the assay conducted during the month/year the column represents. ‘n’ indicates the total number of replicates for each treatment.

Treatments	Jul 2021	Aug 2021	Jun 2022	Jul 2022	n = __
RO water	x	x	x	x	56
Cyantraniliprole	x	x			16
Mineral oil (PSG) 0.25%				x	20
Mineral oil (PSG) 1%	x	x		x	36
SUN-D-06 PS 0.22%			x	x	40
SUN-D-06 PS 0.22% + PSG 0.25%			x	x	40

**Table 3 antibiotics-12-00495-t003:** Treatments evaluated for efficacy against western flower thrips (WFT) in greenhouse assays conducted during 2022. ‘x’ indicates the specific treatment was evaluated during the month/year the column represents. ‘n’ indicates the total number of replicates for each treatment.

Treatments	Jul 2022	Aug 2022	Sep 2022	Oct 2022	n = __
RO water	x	x	x	x	90
Pure Spray Green (PSG) 0.25%	x	x	x	x	90
Pure Spray Green (PSG) 1%			x		30
SUN-D-06 PS 0.22%	x	x	x	x	90
SUN-D-06 PS 0.22% + PSG 0.25%	x	x	x	x	90

## Data Availability

Not applicable.

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
