# Peer review of "Efficacy of Oil and Photosensitizer against Frankliniella occidentalis in Greenhouse Sweet Pepper"

_antibiotics, 2023, doi:10.3390/antibiotics12030495_

Round 1
Reviewer 1 Report
The manuscript entilted “Future IPM: Efficacy of oil and photosensitizer against Frankliniella occidentalis” shows the comparison between a common pesticide used for Western Flower Thrips (WTF), a mineral-oil based spray, and a photosensitizer formulation based on chlorophyllin molecule. The authors tested these two compounds in Brassica oleracea var. Capitata leafs under different situations (minimal light conditions, LED light conditions and in greenhouse conditions), and in many cases the found similar results. It’s very understanding the search for more sustainable and ecologically healthy, and the use of photosensitizer seems to be more indicated than other because it’s not interfered in the plant healthy…
For this referee some points should be discussed or even explained:
1- If the mineral-oil based PSG are approved and has high levels of efficacy in WTF control, why the photosensitizer should be used? This should be very clear in the text. For this referee, there are no plausible justification in the manuscript that become the work very important.
2- How the authors explain the activation of the photosensitizer in the minimal light conditions? In some experiments there are no significative differences with LED ones. This should be discussed in the text.
As minor revision:
· • The authors use acronyms in the title: Future IPM. This should be avoided, please written in a complete way.
· • The description of figure 5 is confused. There are 3 different Figures 5 (Fig. 5A, Fig. 5B and Fig. 5C). The authors should refer the figures correctly.
Author Response
RESPONSE to Reviewer 1
Reviewer 1:
Comments:
The manuscript entilted “Future IPM: Efficacy of oil and photosensitizer against Frankliniella occidentalis” shows the comparison between a common pesticide used for Western Flower Thrips (WTF), a mineral-oil based spray, and a photosensitizer formulation based on chlorophyllin molecule. The authors tested these two compounds in Brassica oleracea var. Capitata leafs under different situations (minimal light conditions, LED light conditions and in greenhouse conditions), and in many cases the found similar results. It’s very understanding the search for more sustainable and ecologically healthy, and the use of photosensitizer seems to be more indicated than other because it’s not interfered in the plant healthy…
For this referee some points should be discussed or even explained:
1- If the mineral-oil based PSG are approved and has high levels of efficacy in WTF control, why the photosensitizer should be used? This should be very clear in the text. For this referee, there are no plausible justification in the manuscript that become the work very important.
2- How the authors explain the activation of the photosensitizer in the minimal light conditions? In some experiments there are no significative differences with LED ones. This should be discussed in the text.
As minor revision:
- The authors use acronyms in the title: Future IPM. This should be avoided, please written in a complete way.
- The description of figure 5 is confused. There are 3 different Figures 5 (Fig. 5A, Fig. 5B and Fig. 5C). The authors should refer the figures correctly.
Reply:
In response to the reviewer’s comment 1, we rearranged and added a few sentences in the introduction to make it clearer that both mineral oil and photosensitizers can fill the need for products that are compatible with IPM programs based on biological control. These two types of products can be either used alone, or in combination (in which case lower concentration of mineral oil can be used to prevent problems such as phytotoxicity).
Comment 2 was addressed by rearranging the paragraph to describe the observed effect (no difference between minimal light and LED), the expected effect (a difference) and the potential explanation (some light was needed for WFT survival and feeding). We also added that “It is possible that the formulated product is more reactive to light under similar conditions, or the light conditions in the growth chamber may have inadvertently changed”.
The title was changed to be more descriptive and avoid the use of the acronym IPM.
Finally, Figure 5 was rearranged to show “total thrips” first followed by “larvae” and “adults”. The graphs were grouped under one figure legend, referring to panel A, B and C. References to the figures were inserted in the text where appropriate.

Reviewer 2 Report
The authors present an interesting study on the control of western flower thrips (WFT; Frankliniella occidentalis) with a novel photosensitized compound. The compound, SUN-D-06 PS, was tested alone or in combination with mineral oil-based spray oil, Pure-17 Spray™ Green (PSG). On the basis of the biological assays (feeding & contact) under various conditions, the results suggest beneficial effects from SUN-D-06 PS. Overall the results and conclusions are well supported by the presented data.
I have made some suggestions below for the authors to consider:
Line 2 - the current title reads more like the tile of a review, I would suggest deleting "Future IPM:". Also noting it is not ideal to have abbreviations in a title.
Line 29 - Suggest adding the species name, Frankliniella occidentalis, to the listed keywords.
Line 57 suggest replacing "enemies" with "predators" or "predators and/or competitors", depending on the intended use of the term enemies - here and elsewhere in the manuscript.
Line 125 suggest abbreviation of "western flower thrips" to "WFT" for consistency.
Line 131 Figure 2 - I am not convinced that this figure is required. Consider removing.
Line 146 & 180 suggest revision "western flower thrips"
Line 148 & 182 suggest revision "that were not significantly different"
Line 217 - I would suggest splitting Figure 5 into separate figures, ie Figure 5A becomes Figure 5, Figure 5B becomes Figure 6 etc.
Line 227 - Figure 5C - what is the rationale for combining larval and adult numbers?
This appears to be to demonstrate some effect related to recruitment, perhaps defined as the number of larvae that survive to become adults.
How were the values calculated that are shown in Figure 5C? Are these the values from the larval and adult counts combined, then averaged based on the number of replicates or the average of the means from each?
Regardless, I am not convinced it is valid to combine these values. The two values are interdependent, as the number of larvae observed, obviously influences the number of resulting adults.
I would suggest the authors consider removing this figure.
Line 261 - Why is there such a broad range in the LC50 estimates?
Line 295-298 - Please remove the statement on unpublished data. The purpose of the discussion is to place the presented results in the context of the existing body of peer-reviewed studies. If the authors believe, their unpublished data is critical to underpin their arguments in this section, it should be added to the manuscript.
Lines 531-539 - Please review and revise this section as no conclusions are presented. There is no scientific reason to mention the manufacturer in the conclusions.
Author Response
RESPONSE to Reviewer 2
Reviewer 2:
Comments:
The authors present an interesting study on the control of western flower thrips (WFT; Frankliniella occidentalis) with a novel photosensitized compound. The compound, SUN-D-06 PS, was tested alone or in combination with mineral oil-based spray oil, Pure-17 Spray™ Green (PSG). On the basis of the biological assays (feeding & contact) under various conditions, the results suggest beneficial effects from SUN-D-06 PS. Overall the results and conclusions are well supported by the presented data.
I have made some suggestions below for the authors to consider:
Line 2 - the current title reads more like the tile of a review, I would suggest deleting "Future IPM:". Also noting it is not ideal to have abbreviations in a title.
Line 29 - Suggest adding the species name, Frankliniella occidentalis, to the listed keywords.
Line 57 suggest replacing "enemies" with "predators" or "predators and/or competitors", depending on the intended use of the term enemies - here and elsewhere in the manuscript.
Line 125 suggest abbreviation of "western flower thrips" to "WFT" for consistency.
Line 131 Figure 2 - I am not convinced that this figure is required. Consider removing.
Line 146 & 180 suggest revision "western flower thrips"
Line 148 & 182 suggest revision "that were not significantly different"
Line 217 - I would suggest splitting Figure 5 into separate figures, ie Figure 5A becomes Figure 5, Figure 5B becomes Figure 6 etc.
Line 227 - Figure 5C - what is the rationale for combining larval and adult numbers? This appears to be to demonstrate some effect related to recruitment, perhaps defined as the number of larvae that survive to become adults.
How were the values calculated that are shown in Figure 5C? Are these the values from the larval and adult counts combined, then averaged based on the number of replicates or the average of the means from each? Regardless, I am not convinced it is valid to combine these values. The two values are interdependent, as the number of larvae observed, obviously influences the number of resulting adults. I would suggest the authors consider removing this figure.
Line 261 - Why is there such a broad range in the LC50 estimates?
Line 295-298 - Please remove the statement on unpublished data. The purpose of the discussion is to place the presented results in the context of the existing body of peer-reviewed studies. If the authors believe, their unpublished data is critical to underpin their arguments in this section, it should be added to the manuscript.
Lines 531-539 - Please review and revise this section as no conclusions are presented. There is no scientific reason to mention the manufacturer in the conclusions.
Reply:
The following changes were made in the same order as the reviewer’s comments:
The title was changed to be more descriptive and avoid the use of the acronym IPM.
The species name, Frankliniella occidentalis, was added to the listed keywords.
In our discipline, “natural enemies” is the correct term to use here. To be more descriptive, we added “(predators)” after “natural enemies” in line 57 and changed “natural enemies” to “biological control agents” in the discussion.
Where appropriate, “western flower thrips” was changed to “WFT”.
We decided to keep figure 2, as it illustrates the photosensitizer compound.
As figure legends should be able to be read apart from the rest of the text, we did not make the change.
As suggested, we changed “do not significantly differ” to "that were not significantly different".
Figure 5 was rearranged to show “total thrips” first followed by “larvae” and “adults”. The graphs were grouped under one figure legend, referring to panel A, B and C. References to the figures were inserted in the text where appropriate. In fact, the total thrips (now panel A) is the sum of larvae (now panel B) and adults (now panel C) in each experimental unit, averaged per treatment. The larvae and adults were presented separately to show (potential) differences in product efficacy against the different thrips stages.
The LC50 values were removed from the discussion to avoid confusion.
As we have no intention of adding or publishing the “unpublished data”, we changed the term to “preliminary data”. The paragraph was rearranged to describe the observed effect (no difference between minimal light and LED), the expected effect (a difference, based on the literature and preliminary trials) and the potential explanation (some light was needed for WFT survival and feeding).
The conclusion was revised to highlight the main result of the study “This study demonstrated that indeed, the photosensitizer, SUN-D-06 PS decreased the WFT population to below threshold in this study on greenhouse pepper”.

Round 2
Reviewer 1 Report
For this referee, at this moment, the manuscript is suitable for publication in tis journal.